# Cross-catenation between position-isomeric metallacages

Yiliang Wang[1], Taotao Liu[1], Yang-Yang Zhang[2], Bin Li[3], Liting Tan[1], Chunju Li [3] ✉, Xing-Can Shen[1] ✉ & Jun Li [2,4]

The study of cross-catenated metallacages, which are complex self-assembly systems arising from multiple supramolecular interactions and hierarchical assembly processes, is currently lacking but could provide facile insights into achieving more precise control over low-symmetry/high-complexity hierarchical assembly systems. Here, we report a cross-catenane formed between two position-isomeric Pt(II) metallacages in the solid state. These two metallacages formed [2]catenanes in solution, whereas a 1:1 mixture selectively formed a cross-catenane in crystals. Varied temperature nuclear magnetic resonance experiments and time-of-flight mass spectra are employed to characterize the cross-catenation in solutions, and the dynamic library of [2] catenanes are shown. Additionally, we searched for the global-minimum structures of three [2]catenanes and re-optimized the low-lying structures using density functional theory calculations. Our results suggest that the binding energy of cross-catenanes is significantly larger than that of self-catenanes within the dynamic library, and the selectivity in crystallization of cross-catenanes is thermodynamic. This study presents a cross-catenated assembly from different metallacages, which may provide a facile insight for the development of low-symmetry/high-complexity self-assemble systems.

Interlocked structures resulting from the self-assembly of biomacromolecules such as DNA[1] and proteins[2] are common in biosystems. Meanwhile, synthetic interlocked supramolecular architectures have been applied to molecular machines and other molecular devices[3–7]. For chemists, coordination-driven self-assembly[8,9] is a promising approach to construct interlocked structures with high synthetic yield[10,11], owing to the involvement of multiple orthogonal supramolecular interactions and hierarchical assembly processes.

Quantitative self-assembled [2]catenane metallacycles were reported by Fujita et al.[12]. In that system, supramolecular interactions (π⋯π interactions and hydrophobic interaction) between metallacycles act as the main energy source to overcome the entropy loss during catenation, giving rise to an equilibrium between monomeric metallacycles and [2]catenanes, while quantitative synthesis of [2] catenane can be achieved in high concentrations. Inspired by this pioneering work, chemists introduced supramolecular interaction binding sites into ligand' framework, combined with metal acceptors to synthesis supramolecular coordination complexes (SCCs). These can further assemble and form a variety of interlocked structures such as rotaxanes[13], molecular knots[14,15], catenanes[16–22], Borromean rings[23,24], interlocked cages[25,26], etc.[27]. However, catenated structures constructed from different SCCs have rarely been reported[28–30], and cross-catenanes assembled from two different metallacages are unknown. The synthetic challenge lies in achieving precise control over the dynamic behavior of SCCs in the hierarchal assembly process. Additionally, relatively low-symmetry final products are difficult to separate and characterize.

[1]State Key Laboratory for Chemistry and Molecular Engineering of Medicinal Resources, Collaborative Innovation Center for Guangxi Ethnic Medicine, School of Chemistry and Pharmaceutical Sciences, Guangxi Normal University, Guilin 541004, PR China. [2]Department of Chemistry, Southern University of Science and Technology, Shenzhen 518005, PR China. [3]Tianjin Key Laboratory of Structure and Performance for Functional Molecules, College of Chemistry, Tianjin Normal University, Tianjin 300387, PR China. [4]Department of Chemistry and Engineering Research Center of Advanced Rare-Earth Materials of Ministry of Education, Tsinghua University, 100084 Beijing, PR China. ✉e-mail: cjli@shu.edu.en; xcshen@mailbox.gxnu.edu.cn

In this study, we successfully synthesized a unique cross-catenane using two positionally isomeric metallacages. The solid-state structure of the cross-catenane was characterized by single-crystal X-ray diffraction, while the dynamic library[31,32] of self-catenated [2]catenanes and cross-catenanes in solution was studied using variable-temperature NMR experiments and time-of-flight mass spectrometry. To gain a deeper understanding of the energetics involved in the catenated assembly, we employed the TGMin (v.3) program[33–36] to search for the global-minimum structures of three [2]catenanes. Density functional theory (DFT) calculations with the B3LYP hybrid exchange-correlation functional[37,38] were used to calculate the binding energies, revealing that the cross-catenane (−77.09 kcal/mol) is significantly more thermodynamically stable than the self-catenated [2]catenanes (−36.79 kcal/mol and −39.49 kcal/mol). Thus, precise control over the crystallization of low-symmetric cross-catenated metallacages is achieved through structural modulations of SCCs in hierarchical self-assembly.

## Results

### Synthesis and structural characterized of metallacages and cross-catenane

A platinum(II) metallacage **4** containing internal supramolecular binding sites that form a [2]catenane in acetone has been previously reported (**4**+**4**, Fig. 1a). A covalently linked bismetallacage was synthesized to form the cyclic bis[2]catenane metallacage[39]. In this study, tweezer-like dipyridine ligand **1**[39], 90° Pt(II) acceptor **2** (*cis*-(PEt₃)₂Pt(OTf)₂, OTf, OSO₂CF₃), and terephthalic acid disodium salt **5** were mixed in acetone/H₂O (v/v = 5:1) to synthesize metallacage **6** (Fig. 1b) through 90° Pt(II) heteroligation[40]. We removed the NaOTf by-

product through extraction[39]. The ¹H and ³¹P{¹H} nuclear magnetic resonance (NMR) spectra (Supplementary Figs. 1 and 2) of the mixture indicate a single, discrete assembly with high symmetry. The ³¹P{¹H} spectrum exhibited two pairs of coupled doublets with chemical shifts at δ = 5.22 and −0.10 ppm, coupling constants $^2J_{P–P}$ = 21.6 Hz, and concomitant ¹⁹⁵Pt satellites were also observed. These suggest the presence of two distinct phosphorus environments, indicating that the Pt(II) center is characterized by a heteroligated coordination moiety consisting of both carboxylate and pyridyl groups, which disrupts the symmetry of the two capping phosphine ligands. Notably, **6** is a positional isomer of **4**, and they share the same internal supramolecular interaction binding sites derived from ligand **1**. However, we expected a slight difference in cavity size between the two metallacages because of the positional isomeric dicarboxylate ligands[41,42]. In this study, structural differences between the two metallacages were minimized by synthetic design. However, the catenation within the mixed system containing these two metallacages poses a significant challenge to studying them due to their similarity. The two positional isomeric metallacages were used to elucidate catenation behavior beyond the skeletons' match[28,43], resulting in a cross-catenane in the solid state (**4**+**6**, Fig. 1b).

To study cross-catenation among these two isomeric metallacages, it is necessary to understand the catenation behaviors of **4** and **6** respectively. Previous reports have described the assembly and single-crystal structure of the [2]catenane **4**+**4** (Fig. 1a), indicating that supramolecular interactions among monomers serve as the primary energy source for overcoming entropy losses during the catenation process[39]. By slowly diffusing isopropyl ether into an acetone solution of **6**, we obtained single crystals of metallacage **6** suitable for X-ray

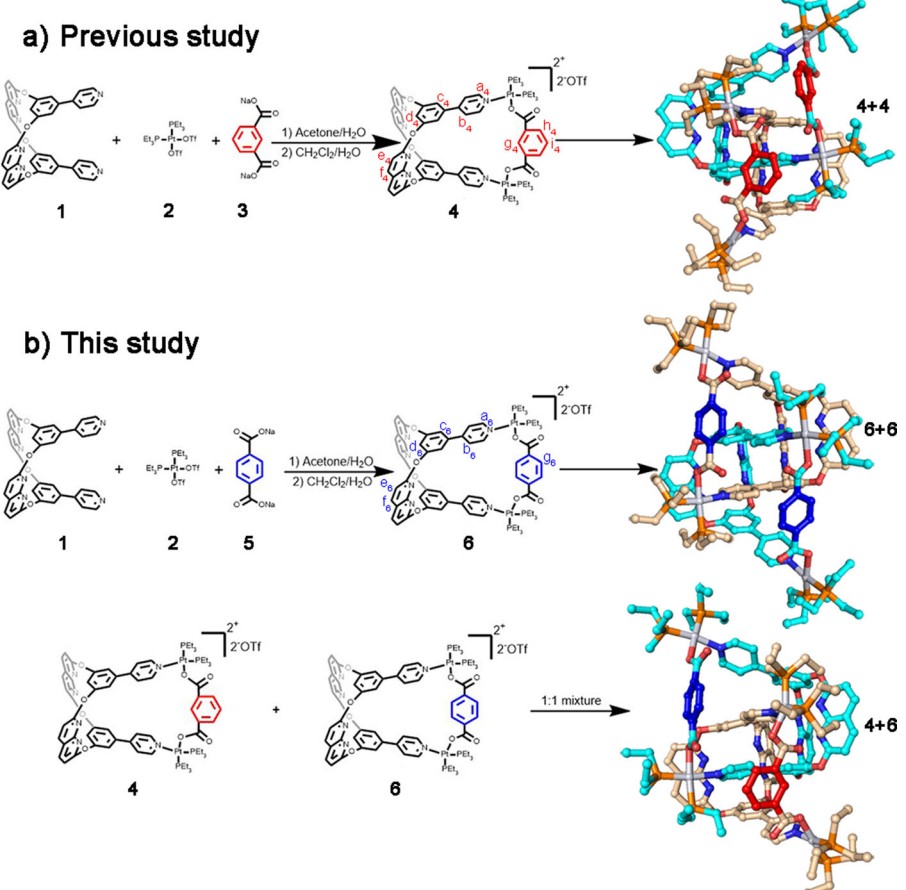

**Fig. 1 | Synthesis and single-crystal structures of metallacages and [2]catenanes. (a)** Synthesis of metallacage **4** and crystal structure of [2]catenane **4**+**4**; (**b**) synthesis of metallacage **6** and crystal structures of [2]catenane **6**+**6** and cross-catenane **4**+**6**. (Hydrogen atoms omitted for clarity).

crystallographic analysis. Additionally, the interlocked structures of two metallacages (Fig. 1b, [2]catenane, **6**+**6**) were observed upon applying symmetry operations. As shown in Supplementary Fig. 7, we observed a slight cavity size difference between **4** and **6** with distances of 8.9 and 9.8 Å, respectively. We define cavity size as the distance between the two N atoms of the pyridine units. We observed multiple supramolecular interactions in the [2]catenane **6**+**6** crystal structure (Supplementary Fig. 8), suggesting that similar forces drove the catenation of **4**+**4** and **6**+**6**. The single-crystal structures of **4**+**4** and **6**+**6** demonstrate similarity in structure and catenation behavior of the two positional isomeric metallacages. Therefore, it is intriguing to investigate the catenated behaviors between these two metallacages. The separation and characterization of the resulting cross-catenane will offer insights into the studies of low-symmetry/high-complexity secondary self-assemble systems. Notably, chemical equilibrium was found to exist in the two metallacages' solutions, and the conversion of [2]catenane **6**+**6** was slightly higher than that of **4**+**4** in the same concentration (20 mM for **4** and **6**, conversion of 74% and 81%, respectively; Supplementary Tables 4 and 5). Moreover, electrospray ionization time-of-flight mass spectrometry (ESI-TOF-MS) data suggested that metallacage **6** and [2]catenane **6**+**6** co-existed in solution (Supplementary Fig. 3), similar to **4**+**4**.

To characterize the cross-catenation between the two metallacages, we dissolved both **4** and **6** at a mole ratio of 1:1 in acetone, and isopropyl ether was slowly added to obtain single crystals. In the crystal structure, **4** and **6** co-existed in the asymmetric unit. After performing a symmetry operation, we observed only cross-catenane **4**+**6** (Fig. 1b). Similar to **4**+**4** and **6**+**6**, we observed multiple intermolecular interactions between **4** and **6** in cross-catenane **4**+**6** (Supplementary Fig. 9), suggesting that the main driving forces of catenation are the C-H···N bonds between benzene-pyridine arms and naphthyridines, and π–π interactions between naphthyridines. Owing to the similarities between the structures of the two positional isomeric metallacages and their catenation behaviors, cross-catenation could not be explained by chiral self-sorting[43] or steric effects[28]. Therefore, this unique result encouraged us to investigate the selectivity in the formation of cross-catenated metallacages.

## NMR experiments and TOF-mass experiments

To verified the generation of cross-catenane in solution, we used [1]H NMR experiments to characterize cross-linking in solution. When the concentration of **4** and **6** in the 1:1 mixture in acetone-$d_6$ solution was gradually increased, several broad peaks were observed from 5.0 to 9.0 ppm in the [1]H NMR spectra. We expected this based on the complex chemical environment that exists in **4**+**6** (Supplementary Fig. 18). Thus, the peaks of cross-catenane **4**+**6** could not be identified by two-dimensional (2D) correlation spectroscopy (COSY) or [1]H diffusion-ordered spectroscopy (DOSY) experiments due to the structural complexity of cross-catenane and similarities between the two isomeric metallacages. However, some identical NMR signals were assigned to **4**+**4** or **6**+**6**, indicating the existence of self-catenation. However, the proportion of **4**+**4** was far lower than that of **6**+**6** within the system based on the integrals of identical [1]H NMR signals (mole ratio of **4**+**4**:**6**+**6** was approximately 1:3, Supplementary Fig. 18). Thus, some metallacage **4** in the 1:1 mixture was trapped in cross-catenane **4**+**6**, whereas some free **6** formed **6**+**6** in the solution. Finally, [1]H NMR spectra indicated the formation of **4**+**6**, but the catenation behavior in solution was random, and the dynamic library was expected.

In order to investigate the stimuli-response of this hierarchal assembled system, we conducted [1]H & [31]P{[1]H} NMR (5 mM for each) spectra in acetone-$d_6$ at different temperatures. As shown in Fig. 2a, the [31]P{[1]H} NMR spectra clearly shown signals belongs to the cross-catenane **4**+**6** at high temperature (pink peaks), while the chemical shifts of **4**+**4** & **6**+**6** are occurred due to the temperature variation. However, as the temperature decreased, the signals of cross-catenane

were disappeared, and the spectra indicated two self-catenanes (**4**+**4** and **6**+**6**) are the main product in the mixture. Additionally, some identical proton signals belonging to the cross-catenane **4**+**6** were clearly visible at high temperatures (Fig. 2b), and chemical shifts can be clearly observed due to the low-symmetric cross-catenated products. Furthermore, at −60 °C, the NMR spectrum reveals that the chemical equilibrium has shifted almost entirely towards the formation of self-catenated states. This suggests that the dynamic library of catenanes is influenced by temperature, indicating that the cross-catenane **4**+**6** is more thermodynamically stable than self-catenanes. That is, heating disrupts the self-catenanes, resulting in the formation of more thermodynamically stable cross-catenanes. Although we were able to identify signals belonging to the cross-catenane **4**+**6**, the low symmetry of the structures made it impossible to calculate the mole ratios of **4**+**6** in the solution based on the integration of the [1]H NMR spectra.

To further describe cross-catenation in solution, metallacage **8** was synthesized based on metallacage **4**. As shown in Fig. 3a, we introduced a methyl group to the metallacage framework through a dicarboxylated ligand, which could provide high-field proton NMR signals for metallacages and catenanes, and extra molecular weight to identify cross-catenanes in mass spectra. The [1]H and [31]P{[1]H} NMR spectra (Supplementary Figs. 4 and 5), as well as the ESI-TOF-MS spectrum (Supplementary Fig. 6), indicate the successful synthesis of metallacage **8**. ESI-TOF-MS also suggested that catenated dimer **8**+**8** existed in the solution of mellatacage **8**, and the results were in good agreement with theory ([M − 3OTf]$^{3+}$, Supplementary Fig. 6). To measure the chemical equilibrium between metallacage **8** and [2]catenane **8**+**8**, we recorded concentration-dependent [31]P{[1]H} NMR & [1]H NMR spectra (Supplementary Figs. 14 and 15). It was seen that methyl's proton signal at δ = 2.17 ppm could be attributed to the **8** monomer, while the signal at δ = 2.68 ppm could be attributed to [2]catenane **8**+**8**. On integrating these two signals, the conversion of **8**+**8** was found to be 75% at a concentration of 20 mM **8** (Supplementary Table 6) in acetone-$d_6$. This is nearly identical to that observed for **4**+**4** in the same concentration.

Encouraged by the identifiable change of methyl in proton NMR signals in the catenated process, we investigated cross-catenation between **6** and **8** in an acetone-$d_6$ solution. As shown in Fig. 3b (entry 3) and Supplementary Fig. 20, when the concentration of the 1:1 mixture of **6** and **8** in acetone-$d_6$ solution was gradually increased, an extra proton signal at δ = 2.60 ppm was clearly visible. Thus, we observed the formation of cross-catenane **6**+**8** in solution, with a conversion of 16.61% at a total concentration of 20 mM (Supplementary Table 7). Moreover, ESI-TOF-MS data also indicated the formation of cross-catenane **6**+**8** in the mixture, and the results were in good agreement with theory ([M − 2OTf]$^{2+}$, Fig. 3d and Supplementary Fig. 22). Additionally, mass signals for [2]catenane **6**+**6** were also observed in the mixture ([M − 2OTf]$^{2+}$, Fig. 3e and Supplementary Fig. 22), and evidence for **8**+**8** [2]catenane was not observed within the mixtures. In summary, we not only characterized the molecular recognition motif between positionally isomeric metallacages in solid-state catenated assemblies by single-crystal X-ray diffraction, but also investigated solution phase cross-catenation by slightly modifying the metallacage frameworks.

## GM structures searching and DFT caculation

To evaluate the stabilities and mechanisms of cross-catenation, TGMin (v.3) program[33–35] based on the basin-hopping algorithm interfaced with the GFN2-xTB package[36] was employed to search the GM structures of three catananes, and the generalized Born and surface area solvation (GBSA) model[44] with acetone as solvent was also taken into consideration. In total, 67320 structural isomers of **4**+**4**, 54589 isomers of **4**+**6**, and 55020 isomers of **6**+**6** were searched, respectively, among all the possible structures. The as-found low-lying structures are

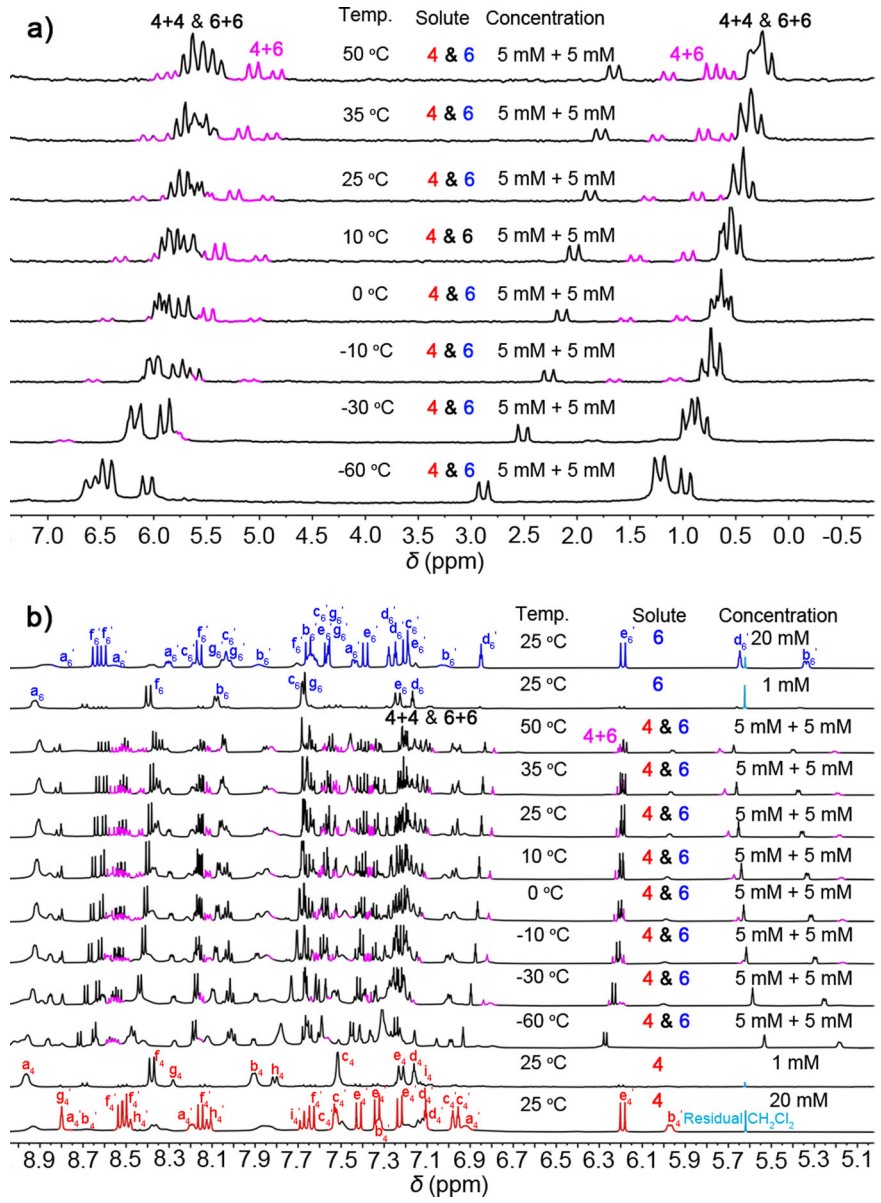

**Fig. 2 | Varied temperature NMR experiments.** (**a**) $^{31}P\{^{1}H\}$ NMR spectra (160 MHz) of **4** and **6** 1:1 mixture in acetone-$d_6$ in different conditions; (**b**) $^{1}H$ NMR spectra (400 MHz) of **4** and **6** mixtures in acetone-$d_6$ in different conditions.

showing in Supplementary Figs. 23–31. Subsequently, the GM structures of **4**+**4, 4**+**6** and **6**+**6** were re-optimized by using B3LYP exchange-correlation functional[37,38] and def2-TZVP basis sets[37] in Gaussian (version G16RevB.01) program[45].

In the GM structures, the complicated hydrogen bonding network consist of C-H···N and C-H···O bonds help to stabilize the geometrical structures of [2]catenanes **4**+**4, 4**+**6** and **6**+**6**, which are observed in crystal structures. Although the geometrical structures of **4**+**4, 4**+**6** and **6**+**6** with cross-catenation between two positional isomeric metallacages are quite similar, the topological structures of them are rather different. The topological structures of **4**+**4** and **4**+**6** are more or less similar, while the topological structure of **6**+**6** is different due to the altered hydrogen bonding network (Fig. 4). One can conclude that the driving force of the cross-catenation between two positional isomeric metallacages is to form the framework of hydrogen bonding networks consisting of stronger C-H···N and C-H···O hydrogen bonds. The cross-catenation structure of [2]catenane prefers the structure with the most stable hydrogen bonding networks.

To account for the experimental results that only **4**+**6** exists in the mixed single crystals, we have calculated the binding energy (BDE) of [2]catenanes **4**+**4, 4**+**6** and **6**+**6** from two respective monomers. As demonstrated in Table 1, the stabilization energies due to dimerization lie in the order of **4**+**6** » **6**+**6** > **4**+**4**. The significantly large BDE of **4**+**6** indicates the mixed single crystals prefer formation of this **4**+**6** structure instead of the **4**+**4** and **6**+**6** structures. Meanwhile, the mathematical probability of the cross-catenation state (2/3) in a mixed system containing two metallacage **4** and two metallacage **6** is higher than that for the self-catenation state (1/3), indicating that the formation of cross-catenane **4**+**6** is entropically favorable in crystallization. This is because the minimal structural differences within the metallacages' frameworks strongly influence the hierarchical catenated assembly. Consequently, unique cross-catenated metallacages are preferentially formed in the crystal.

## Discussion
In conclusion, we reported the unique example of a cross-catenane obtained from positional isomeric metallacages with single-crystal

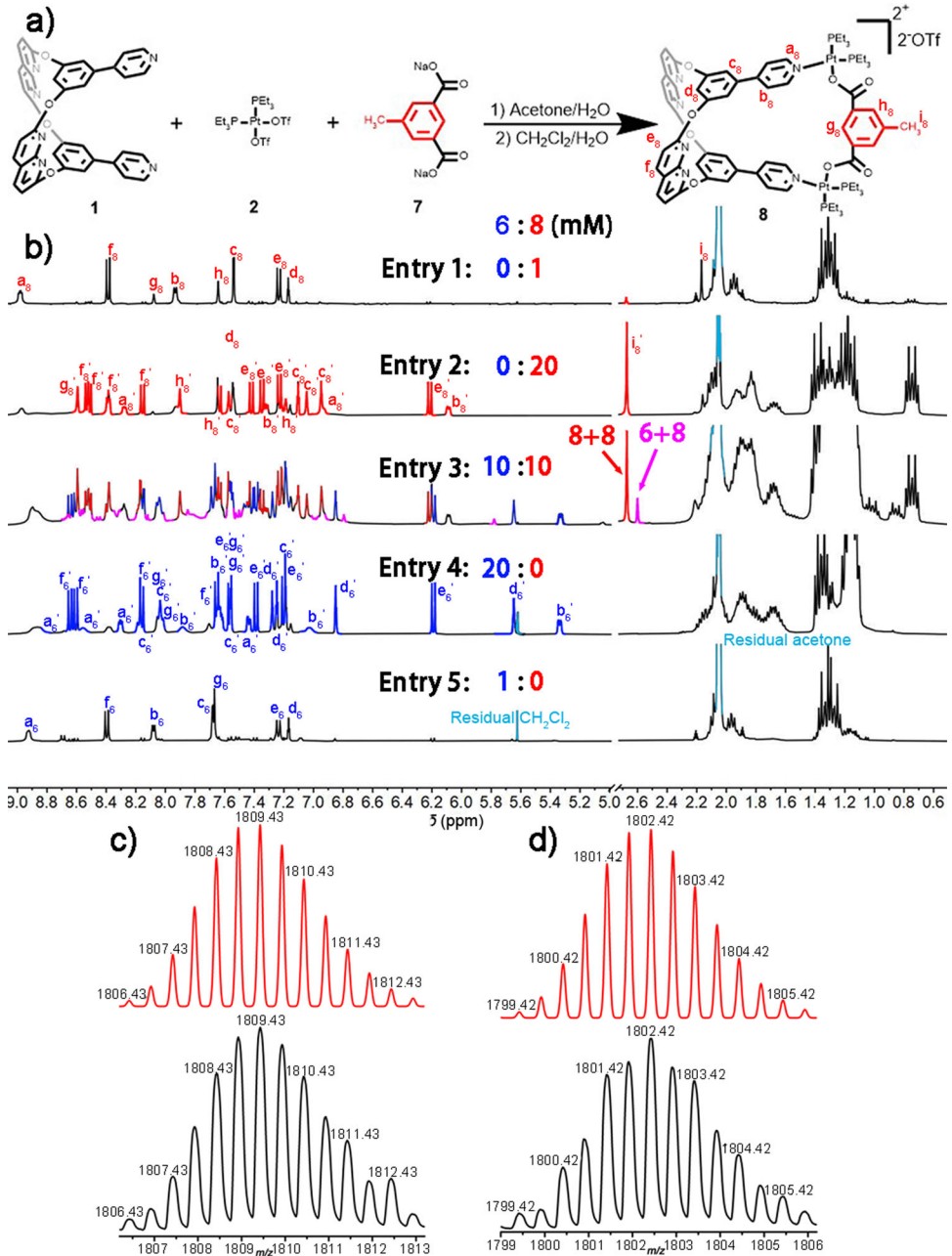

**Fig. 3 | Synthesis and structural characterize of metallacage 8 and cross-catenane 6+8. (a)** Synthesis of metallacage **8**; **(b)** [1]H NMR spectra (400 MHz, 298 K) of **6** and **8** in acetone-$d_6$ in different concentrations; **(c)** experimental (black) and calculated (red) ESI-TOF-MS spectra of **6+8** cross-catenane ([M-2OTf][2+]) & **(d)** experimental (black) and calculated (red) ESI-TOF-MS spectra of **6+6** [2]catenane ([M-2OTf][2+]) in acetone solution of **6** & **8** 1:1 mixture.

structures. The solid-state structure of the cross-catenane was characterized by single-crystal XRD, and the dynamic library of catenanes in solution was characterized by NMR and TOF-MS spectra. We conducted global-minimum structure searching and DFT calculations to elucidate the formation mechanism, revealing that the binding energy of the cross-catenane is significantly larger than that of self-catenanes, and thermodynamic driving forces mainly control the selectivity in crystallization. Our findings demonstrate that minimizing structural modulation of SCCs can precisely control their hierarchical interlocked behaviors during crystallization. This result offers facile insight into the future development of low-symmetry/high-complexity supramolecular interlock systems. Further investigations of additional examples are currently underway.

## Methods
All reagents were commercially available and used as supplied without further purification. Deuterated solvents were purchased from Cambridge Isotope Laboratory (Andover, MA). Compound **1**[39], **2**[40], **4**[39], **5**[40] was prepared according to the published procedures. [1]H NMR, [31]P{[1]H} NMR spectra and 2D COSY NMR spectra were recorded on Bruker AVANCE III HD 400 MHz spectrometer and Bruker AVANCE III HD 600 MHz spectrometer. [1]H NMR chemical shifts are reported relative to residual solvent signals. Mass spectra were recorded on the Micromass Quattro II triple-quadrupole mass spectrometer using electrospray ionization, Thermo Scientific Q Exactive mass spectrometer using electrospray ionization and Agilent 6545 Q-TOF using electrospray ionization. The single crystals data were collected on a Nonius

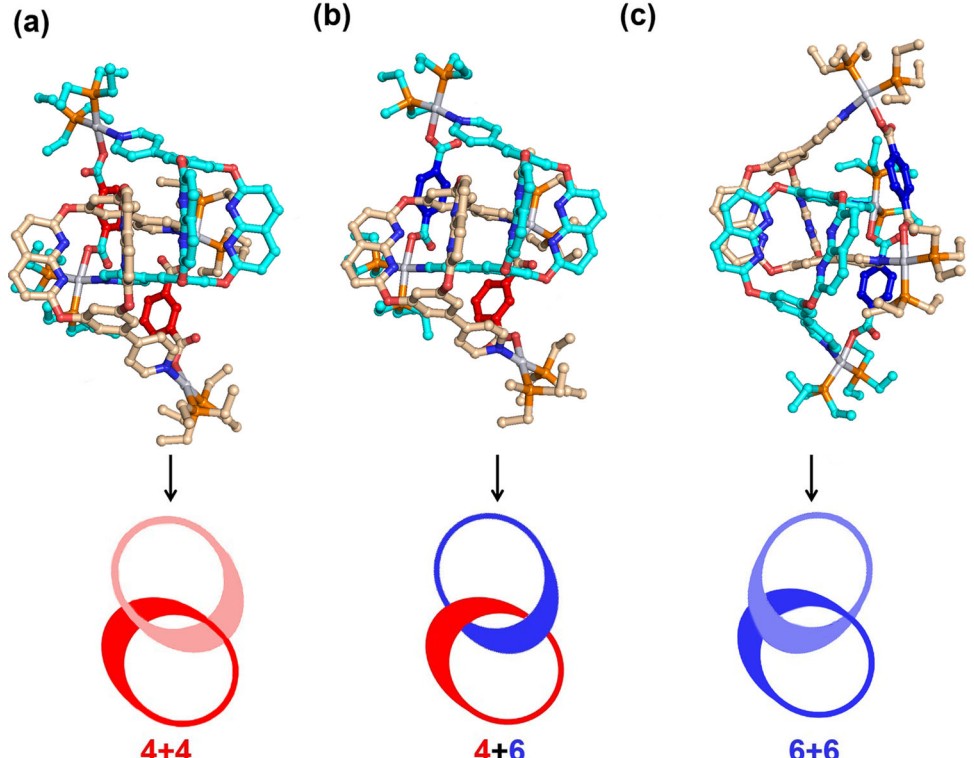

**Fig. 4 | The global-minimum geometrical structures of [2]catenanes. (a) 4+4; (b) 4+6** and **(c) 6+6** optimized at the B3LYP/def2-TZVP level of theory.

KappaCCD diffractometer equipped with Mo K-alpha radiation ($\lambda = 0.71073$ Å) and a BRUKER APEXII CCD.

## Synthesis of 6

**1** (6.27 mg, 10 μmol), *cis*-Pt(PEt₃)₂(OTf)₂ **2** (14.59 mg, 20 μmol) and dicarboxylate ligand **5** (2.10 mg, 10 μmol), were placed in a 2-dram vial, followed by addition of $H_2O$ (0.2 mL) and acetone (1.0 mL). After 3 h of heating at 50 °C, all solvent was removed by $N_2$ flow. Acetone (1.0 mL) was then added into the resultant mixture, and the solution was stirred at 50 °C for 8 h. The resulting product was precipitated with diethyl ether, isolated and dried under reduced pressure and dissolved in $CH_2Cl_2$ (3.0 mL). Above solution was placed in a 2-dram vial, followed by addition of $H_2O$ (2.0 mL), and stirred at room temperature for 12 h. Organic phase was collected and dried over anhydrous MgSO₄, filtered. $CH_2Cl_2$ was removed by $N_2$ flow and the solid was dried under vacuum, white solid: 18.62 mg, Yield: 95%. The ¹H NMR spectrum of **6** is shown in Supplementary Fig. 1. ¹H NMR (400 MHz, acetone-$d_6$, 298 K) δ (ppm): 8.92 (m, 4H), 8.40 (d, $J = 8.7$ Hz, 4H), 8.08 (d, $J = 6.1$ Hz, 4H), 7.68 (d, $J = 2.1$ Hz, 4H), 7.67 (s, 4H), 7.24 (d, $J = 8.6$ Hz, 4H), 7.17 (t, $J = 2.0$ Hz, 2H), 2.12 – 2.04 (m, 48H), 1.41-1.20 (m, 72H). The ³¹P{¹H} NMR spectrum of **6** is shown in Supplementary Fig. 2. ³¹P{¹H} NMR (160 MHz, acetone-$d_6$, 298 K) δ (ppm): 5.22 (d, $^2J_{\text{P-P}} = 21.8$ Hz, ¹⁹⁵Pt satellites, $^1J_{\text{Pt-P}} = 3187$ Hz), and −0.10 (d, $^2J_{\text{P-P}} = 21.6$ Hz, ¹⁹⁵Pt satellites, $^1J_{\text{Pt-P}} = 3372$ Hz). ESI-TOF-MS is shown in Supplementary Fig. 3: 826.24 [M − 2OTf]²⁺.

## Synthesis of 8

**1** (6.27 mg, 10 μmol), *cis*-Pt(PEt₃)₂(OTf)₂ **2** (14.59 mg, 20 μmol) and dicarboxylate ligand **7** (2.24 mg, 10 μmol), were placed in a 2-dram vial, followed by addition of $H_2O$ (0.2 mL) and acetone (1.0 mL). After 3 h of heating at 50 °C, all solvent was removed by $N_2$ flow. Acetone (1.0 mL) was then added into the resultant mixture, and the solution was stirred at 50 °C for 8 h. The resulting product was precipitated with diethyl ether, isolated and dried under reduced pressure and dissolved in $CH_2Cl_2$ (3.0 mL). Above solution was placed in a 2-dram vial, followed by addition of $H_2O$ (2.0 mL), and stirred at room temperature for 12 h. Organic phase was collected and dried over anhydrous MgSO₄, filtered. $CH_2Cl_2$ was removed by $N_2$ flow and the solid was dried under vacuum, white solid: 18.28 mg, Yield: 93%. The ¹H NMR spectrum of **8** is shown in Supplementary Fig. 4. ¹H NMR (400 MHz, CD₂Cl₂, 298 K) δ (ppm): 8.77 (m, 4H), 8.13 (d, $J = 8.6$ Hz, 4H), 7.90 (s, 1H), 7.74 (m, 4H), 7.53 (d, $J = 1.6$ Hz, 2H), 7.36 (d, $J = 2.1$ Hz, 4H), 7.14 (d, $J = 8.6$ Hz, 4H), 7.11 (t, $J = 2.0$ Hz, 2H), 2.11 (s, 3H), 1.92 – 1.67 (m, 48H), 1.30 – 1.17 (m, 72H). The ³¹P{¹H} NMR spectrum of **6** is shown in Supplementary Fig. 5. ³¹P{¹H} NMR (160 MHz, CD₂Cl₂, 298 K) δ (ppm): 4.98 (d, $^2J_{\text{P-P}} = 21.5$ Hz, ¹⁹⁵Pt satellites, $^1J_{\text{Pt-P}} = 3190$ Hz), and −0.77 (d, $^2J_{\text{P-P}} = 21.6$ Hz, ¹⁹⁵Pt satellites, $^1J_{\text{Pt-P}} = 3380$ Hz). ESI-TOF-MS is shown in Supplementary Fig. 6: 833.24 [M − 2OTf]²⁺; 1815.43 [M − OTf]⁺.

## Crystal structure determination and refinements

Single crystals of **6+6** and **4+6** were obtained by slow diffusion of isopropyl ether into their acetone solution. Single-crystal X-ray diffraction data was collected on a Nonius KappaCCD diffractometer equipped with Mo K-alpha radiation ($\lambda = 0.71073$ Å) and a BRUKER APEXII CCD. Throughout data collection, the crystal was cooled with an Oxford Cryosystem. The crystal structure was solved and refined against all $F^2$ values using the SHELX and Olex 2 suite of programmes[46,47]. Single crystal of [2]catenane **6+6** and **4+6** presents large voids filled with a lot of scattered electron density. Solvent mask protocol inside Olex 2 software was used to account for the void electron density corresponding to the disordered OTf⁻ anions and solvent molecules placed in the intermolecular space in the crystal structure. Hydrogen atoms were not placed in some PEt₃ groups' calculated positions to avoid disorder.

A large number of A-alerts and B-alerts were found due to poor resolution because of the low qualities of single-crystals and large asymmetry units; thus responses of alerts are shown in Supplementary Table 1.

**Table 1 | The calculated binding energy (BDE, in kcal/mol) of [2]catenanes 4+4, 4+6 and 6+6 from two monomers calculated at the DFT B3LYP/def2-TZVP level of theory**

| Compound | Reaction | BDE | DBDE |
|---|---|---|---|
| 4+4 | 4+4 → 4+4 | −36.79 | 40.3 |
| 4+6 | 4+6 → 4+6 | −77.09 | 0 |
| 6+6 | 6+6 → 6+6 | −39.49 | 37.6 |

## GM structures searching and DFT caculation

The global-minimum (GM) structures of [2]catenanes **4**+**4**, **4**+**6** and **6**+**6** were searched using TGMin (v.3) program[33–35] based on the basin-hopping algorithm interfaced with the GFN2-xTB package[36], respectively. The generalized Born and surface area solvation (GBSA) model[44] with acetone as solvent was also taken into consideration. In total, 67320 structural isomers of **4**+**4**, 54589 isomers of **4**+**6**, and 55020 isomers of **6**+**6** were searched, respectively, among all the possible structures. The as-found low-lying structures are showing in Supplementary Figs. 25–31.

Based the above-located low-lying structures, the **4**+**4**, **4**+**6** and **6**+**6** GM structures were re-optimized by using density functional theory (DFT) with B3LYP hybrid exchange-correlation functional[37,38] and def2-TZVP basis sets[48] in Gaussian (version G16RevB.01) program[44]. Given the van der Waals interaction in these systems, Grimme's D3 correction was also applied to take the dispersion correction into account. Furthermore, the universal solvation model based on density (SMD)[49,50] with acetone as solvent was also applied in the DFT calculations in order to account for the solvent effect. The binding energies (BDE) were calculated following the thermodynamic convention: $BDE = \sum E_{complex} - \sum_i(E_{fragment(i)})$, where $E_{complex}$ and $E_{fragment(i)}$ are the energies of the complex and the $i$-th constituent fragment, respectively. The calculated DFT results are shown in Fig. 4 and Supplementary Figs. 23 and 24.

## Data availability

The X-ray crystallographic coordinates for structures reported in this study have been deposited at the Cambridge Crystallographic Data Center (CCDC), under deposition numbers 2178656 (metallacage **6** & [2]catenane **6**+**6**) and 2178655 (cross-catenane **4**+**6**). These data can be obtained free of charge from The Cambridge Crystallographic Data Center via www.ccdc.cam.ac.uk/data_request/cif. The source data of the coordinates of the optimized structures are present. All additional data are available from the corresponding author upon request. Source data are provided with this paper.

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

## Acknowledgements

This work was supported by National Natural Science Foundation of China (22377018, X.-C. S. and 22301050, Y. W.), China Postdoctoral Science Foundation (2021M690766, Y. W.), Natural Science Foundation of Guangxi Province (23GXNSFBA026318, Y. W.), the BAGUI Scholar Program (X.-C. S.), and the State Key Laboratory Cultivation Base for Chemistry, Molecular Engineering of Medicinal Resources (CMEMR 2021-A02, Y. W.).

## Author contributions

Y. W., C. L. and X.-C. S. conceived and designed the experiments. T. L., Y. W., B. L. and X. Z. performed the experiments. Y.-Y. Z. and J. L. performed the search for the global-minimum structures and DFT calculations. T. L., Y. W., Y.-Y. Z., L. T. and B. L. analyzed the results and wrote the paper. X.-C. S. supervised the entire project.

## Competing interests

The authors declare no competing interests.
