## [Peer Review File · Nature Communications]

Reviewers' Comments:

Reviewer #1:

Remarks to the Author:

Shen and co-workers report the synthesis of two cross-[2]catenane structures based on the previous work, characterizing the structures in the solid state by X-ray crystallographic means and using liquid NMR hydrogen spectroscopy as well as high-resolution mass spectrometry for an in-depth study of the structure's behavior in solution; in addition, the authors use theoretical calculations to provide additional evidence.

Overall, the core content of this work is innovative and helpful in understanding the formation of this heterogeneous metalla-[2]catenane structures. The research approach to catenane formation in this work is also a good reference case, but as far as the current manuscript is concerned, it needs further improvement, especially in the details, which are suggested as follows:

1) The question of whether reference 31, cited by the authors in the text, should be reference 37, the paper published by Prof. Stang's group in 2020 (Nat Commun, 2020, 11, 2727), is not Nano Res. 2017, 10, 3407-3420 cited in the manuscript, and I noted that the current manuscript cites a paper on theoretical computational literature.

2) Regarding the presentation of NMR spectra, the symbols indicating chemical shifts δ should be italicized, and it is also suggested that the authors should label the peak signals of deuterated solvent when presenting the results of the ^1H NMR characterization, so that the reader can easily understand the detailed parts.

3) As shown in Figure 3a in the manuscript, the structure of compound 6 undergoes a monocyclic to [2]catenane conversion along with a change in concentration, and in the supporting information, the ^1H NMR of compound 6 should be indicated with the corresponding concentration, which I understand to be at a low concentration, and similarly, the concentration should be indicated for the characterization of compound 8.

4) I noticed that the deuterated reagent used in compound 6 and the subsequent mixed experiments of compounds 6 and 8 was acetone, but the deuterated solvent used for compound 8 alone was dichloromethane, and I'm curious as to why that was done. I would like to understand the reasons for this.

5) For Figure 3e, it is strongly recommended that the mass spectral characterization data on 6+6 be re-checked by the authors, as it is highly probable that there is a superposition of its peaks with those of other ions. Their relative abundance differs significantly from the calculated value.

6) For the structural characterization of [2]catenanes in solution, I believe DOSY should be supplemented to determine if only monocyclic as well as [2] catenane structure are present in the system, while characterization tests of the NOESY 2D NMR spectra could provide additional support in characterizing the associated hydrogen signals.

7) The authors used theoretical calculations in the article to gain insight into the formation of this type of cross-[2]catenane structures. However, the relevant theoretical calculations parameters were not mentioned in the supporting information, only the results of the calculations. I believe this section should be added to the supporting information to help readers better understand the work.

Reviewer #2:

Remarks to the Author:

In this work, Wang et al describe the cross catenation of metallacages with different chemical structures. Cages that are positional isomers interlock to produce catenanes with two different components in the solution and solid states. This work represents an interesting discovery; I can see also how perhaps equipping the different components with e.g. catalytic units could be useful for applying these catenanes in a future work. However, there are several questions for me that the authors need to address before this work is, in my opinion, publishable. These are:

- How novel really is this work? The authors themselves describe how they used compounds that are either identical or very similar to those published before: 'In this study, structural differences between

the two metallacages were minimised by synthetic design'; 'The single-crystal structures of 4+4 [known from previous work] and 6+6 demonstrate similarity in structure and catenation behaviour of the two positional isomeric metallacages.' I think the novelty could really be explained better if the authors clearly explain the parameters that favour or control cross-catenation. Why, for example, does the cross-catenane crystallise selectively? Why is cross catenation favoured more at elevated temperature? Otherwise, the result is a more of a 'quirk' – a similar result could arguably be obtained by simply isotopically labelling a carbon atom in half of a sample of 4, call it 4' and then claim that 4+4' is cross catenation.

- The authors report various 'yields'. Given the catenanes were not isolated in these cases, this term is not correct. Determining a ratio would be more correct. Also, presenting yields to 4 significant figures (e.g. 74.08%) is not appropriate – NMR integration is not this precise.
- What evidence do the authors have that the catenanes are the same structures in solution and the solid state? Are the relative 3D arrangements of the components the same (i.e. are components threaded through the same component 'entrances'? Given that the constituent cages have three 'entrances' to their cavities, there are several ways they could interlock to form a catenane (I think different threading would produce topological isomers).
- The authors need to assign and explain changes in the $^1\text{H}/^{31}\text{P}$ NMR for the cross-catenanes vs the self catenanes – why do changes in chemical shifts occur?
- Some figures are not well presented:

In figure 1, the crystal structures are hard for me to interpret (they are, in fact, better in the SI – figures S8 and S9 are really nice!).

In Figure 2. Why is the 9 from 8.9 a different font to the rest of the numbers on the NMR scale?

Figure 4: Again, the crystal structures are near impossible to interpret. Also the use of cartoons that show non-interlocked rings is confusing and a little sloppy.

- The authors state: 'The cross-catenane 4+6 was thought to be the most thermodynamically favorable assembly among 4+4, 6+6, and 4+6, and the crystallization process was thermodynamically stable.' Clearly, this statement is not true for all conditions. Generally, the amount ('yield') of cross-catenane formed was small (and the reactions are under thermodynamic control). Please can the authors clarify what they mean. A discussion of entropy is also needed – presumably cross-catenation is entropically favoured (and hence why more cross-catenanes forms at higher temperatures).
- The authors state 'evidence for 8+8 [2]catenane was not observed' but also present a well matched isotope pattern for 8+8 in figure 3c. This is confusing/contradictory – what are the authors trying to say?

Reviewer #3:

Remarks to the Author:

The paper is well written and the results are properly rationalised and justified.

The work represents a nice example of synergic usage of experiments and modelling techniques in the formation of supramolecular metallocage systems.

The work is of good significance in the the field and experimental and computational techniques under use are up to date.

The work is original: it contains new metallocage systems.

The work support the conclusions.

Data data analysis, interpretation and conclusions are reasonable.

The applied methodology is appropriate and respects the standards in the field.

The details provided are enough to reproduce the results.

I suggest to furnish also the geometries of the optimised assemblies obtained by DFT calculations.

I wish to suggest further references to the same subject:

J Martí-Rujas, S Ma, A Famulari - Inorganic Chemistry, 2022

J Martí-Rujas, S Elli, A Famulari - Scientific Reports, 2023

Reply to Reviewer #1

Comments: Shen and co-workers report the synthesis of two cross-[2]catenane structures based on the previous work, characterizing the structures in the solid state by X-ray crystallographic means and using liquid NMR hydrogen spectroscopy as well as high-resolution mass spectrometry for an in-depth study of the structure's behavior in solution; in addition, the authors use theoretical calculations to provide additional evidence.

Reply: Thanks a lot for reviewer's positive comments.

Overall, the core content of this work is innovative and helpful in understanding the formation of this heterogeneous metalla-[2]catenane structures. The research approach to catenane formation in this work is also a good reference case, but as far as the current manuscript is concerned, it needs further improvement, especially in the details, which are suggested as follows:

Reply: We greatly appreciate the reviewer's very positive comments and useful advices. Herewith, we addressed the reviewer's comments as follows:

1) The question of whether reference 31, cited by the authors in the text, should be reference 37, the paper published by Prof. Stang's group in 2020 (Nat. Commun, 2020, 11, 2727), is not Nano Res. 2017, 10, 3407-3420 cited in the manuscript, and I noted that the current manuscript cites a paper on theoretical computational literature.

Reply: Sorry to have caused confusion. We have carefully checked all the cited references and cross-citations in the manuscript. The reference "Nat. Commun. 2020, 11: 2727" has been cited as reference No. 39.

2) Regarding the presentation of NMR spectra, the symbols indicating chemical shifts δ should be italicized, and it is also suggested that the authors should label the peak signals of deuterated solvent when presenting the results of the ^1H NMR characterization, so that the reader can easily understand the detailed parts.

Reply: We appreciate your useful comment. The NMR spectra have been updated to italicize all chemical shifts δ in both the manuscript and

supplementary information. Additionally, residual deuterated acetone and dichloromethane from the metallacage synthesis reactions have been clearly labeled in the spectra.

3) As shown in Figure 3a in the manuscript, the structure of compound **6** undergoes a monocyclic to [2]catenane conversion along with a change in concentration, and in the supporting information, the ^1H NMR of compound **6** should be indicated with the corresponding concentration, which I understand to be at a low concentration, and similarly, the concentration should be indicated for the characterization of compound **8**.

Reply: We appreciate your valuable advice. The legends of Supplementary Figures 1, 2, and 5 have been updated to include the concentration information for **6** and **8**. For **8**, catenation is prohibited in CD_2Cl_2 , hence the legend of Supplementary Figure 4 remains unchanged.

4) I noticed that the deuterated reagent used in compound **6** and the subsequent mixed experiments of compounds **6** and **8** was acetone, but the deuterated solvent used for compound **8** alone was dichloromethane, and I'm curious as to why that was done. I would like to understand the reasons for this.

Reply: We apologize for any confusion caused to the reviewer. Due to the possibility of overlap between the methyl proton signals of **8** and the residual solvent signals of deuterated acetone at high-field, the ^1H NMR spectrum of **8** alone was recorded in deuterated dichloromethane to confirm the successful synthesis of metallacage **8**, as shown in Supplementary Figure 4.

5) For Figure 3e, it is strongly recommended that the mass spectral characterization data on **6+6** be re-checked by the authors, as it is highly probable that there is a superposition of its peaks with those of other ions. Their relative abundance differs significantly from the calculated value.

Reply: We appreciate the reviewer's comment. We have retested the mass spectra for the 1:1 mixtures of **6** & **8** in acetone using an Agilent 6545 Q-TOF with ESI. The formation of cross-catenane **6+8** has been verified and is in good agreement with theory ($[\text{M} - 2\text{OTf}]^{2+}$). Additionally, the mass signals for

[2]catenane **6+6** are in agreement with theory ($[M - 2OTf]^{2+}$) and the isotopic peaks exhibit a Gaussian distribution (Figure 3 in the manuscript and Supplementary Figure 22).

Supplementary Figure 22. Experimental (black) and calculated (red) electrospray ionization mass spectrum of 1:1 mixture of **6** & **8**.

Figure 3 | Synthesis and structural characterization of metallacage **8 and cross-catenane **6+8**.** (a) Synthesis of metallacage **8** (b) ¹H NMR spectra (400 MHz, 298 K) of **6** and **8** in acetone-*d*₆ in different concentrations (c) experimental (black) and calculated (red) ESI-TOF-MS spectra of **6+8** cross-catenane ([M-2OTf]²⁺) (d) experimental (black) and calculated (red) ESI-TOF-MS spectra of **6+6** [2]catenane ([M-2OTf]²⁺).

6) For the structural characterization of [2]catenanes in solution, I believe DOSY should be supplemented to determine if only monocyclic as well as [2]catenane structure are present in the system, while characterization tests of the NOESY 2D NMR spectra could provide additional support in characterizing the associated hydrogen signals.

Reply: We appreciate your useful comment. We have tested the DOSY spectra for **6** and **8** at 10 mM in acetone- d_6 , in which both metallacages and [2]catenanes co-exist in the systems. For metallacage **6** and [2]catenane **6+6**, the DOSY spectrum can separate them to some extent and identify the monomer metallacages as well as [2]catenanes (Figure R1). However, under certain conditions, metallacage **8** and [2]catenane **8+8** cannot be fully resolved (Figure R2). Due to limitations in our NMR probes, we are unable to efficiently separate metallacages and [2]catenanes, and as a result, we cannot provide 2D DOSY spectra in the manuscripts or supplementary information.

Figure R1. 2D DOSY NMR spectra (400 MHz, acetone- d_6 , 298 K, 10 mM for **6**) of **6** & **6+6**.

Figure R2. 2D DOSY NMR spectra (400 MHz, acetone- d_6 , 298 K, 10 mM for **8**) of **8** & **8+8**.

7) The authors used theoretical calculations in the article to gain insight into the formation of this type of cross-[2]catenane structures. However, the relevant theoretical calculation parameters were not mentioned in the supporting information, only the results of the calculations. I believe this section should be added to the supporting information to help readers better understand the work.

Reply: Thank you for your comments. The relevant theoretical calculation parameters have been included in the supporting information of the updated version.

Reply to Reviewer #2

Comments: In this work, Wang et al describe the cross catenation of metallacages with different chemical structures. Cages that are positional isomers interlock to produce catenanes with two different components in the solution and solid states. This work represents an interesting discovery; I can see also how perhaps equipping the different components with e.g. catalytic units could be useful for applying these catenanes in a future work. However, there are several questions for me that the authors need to address before this

work is, in my opinion, publishable. These are:

Reply: We greatly appreciate the reviewer's very positive comments and useful advices. Herewith, we addressed the reviewer's comments as follows:

- How novel really is this work? The authors themselves describe how they used compounds that are either identical or very similar to those published before: 'In this study, structural differences between the two metallacages were minimised by synthetic design'; 'The single-crystal structures of 4+4 [known from previous work] and 6+6 demonstrate similarity in structure and catenation behaviour of the two positional isomeric metallacages.' I think the novelty could really be explained better if the authors clearly explain the parameters that favour or control cross-catenation. Why, for example, does the cross-catenane crystallise selectively? Why is cross catenation favoured more at elevated temperature? Otherwise, the result is a more of a 'quirk' – a similar result could arguably be obtained by simply isotopically labelling a carbon atom in half of a sample of 4, call it 4' and then claim that 4+4' is cross catenation.

Reply: Thank you for your questions, and we apologize for any confusion regarding the novelty of our manuscript. We have made the following modifications to the manuscript: "In this study, structural differences between the two metallacages were minimized by synthetic design." has been revised to read: "In this study, structural differences between the two metallacages were minimized by synthetic design. However, the catenation within the mixed system containing these two metallacages poses a significant challenge to studying them due to their similarity." We hope this clarifies the matter for you.

The words "The single-crystal structures of **4+4** and **6+6** demonstrate similarity in structure and catenation behaviour of the two positional isomeric metallacages. This provides insight into cross-catenation between two different metallacages, extending beyond the skeletons' match in the catenation processes." have been modified to read: "The single-crystal structures of 4+4 and 6+6 demonstrate similarity in structure and catenation behaviour of the two positional isomeric metallacages. Therefore, it is intriguing to investigate the

catenated behaviours between these two metallacages. The separation and characterization of the resulting cross-catenane will offer insights into the studies of low-symmetry/high-complexity secondary self-assemble systems."

The revised text provides a more detailed and accurate description of the similarity in structure and catenation behaviour of the metallacages, as well as the importance of studying their catenated behaviours. It also emphasizes the challenges and intricacies associated with low-symmetry/high-complexity secondary self-assemble systems.

To further clarify the selectivity in crystallization, we have added additional explanations: "Meanwhile, the mathematical probability of the cross-catenation state (2/3) in a mixed system containing two metallacage **4** and two metallacage **6** is higher than that for the self-catenation state (1/3), indicating that the formation of cross-catenane **4+6** is entropically favorable in crystallization. This is because the minimal structural differences within the metallacages' frameworks strongly influence the hierarchical catenated assembly. Consequently, unique cross-catenated metallacages are preferentially formed in the crystal."

Also, the explanations "Furthermore, at -60 °C, the NMR spectrum reveals that the chemical equilibrium has shifted almost entirely towards the formation of self-catenated states. This suggests that the dynamic library of catenanes is influenced by temperature, indicating that the cross-catenane **4+6** is more thermodynamically stable than self-catenanes." have been added.

In summary, this study not only describes the fascinating structure of the cross-catenane containing two positional isomeric metallacages, but also offers valuable insights into the processes of separation, characterization, and formation mechanisms of low-symmetry/high-complexity self-assemble systems. We hope these clarifies the matter for you.

- The authors report various 'yields'. Given the catenanes were not isolated in these cases, this term is not correct. Determining a ratio would be more correct. Also, presenting yields to 4 significant figures (e.g. 74.08%) is not

appropriate – NMR integration is not this precise.

Reply: Thank you for your comments. The term "yields" in the manuscript and supplementary information has been replaced with "conversion". Additionally, the conversion of [2]catenanes has been adjusted (e.g., 74.08% was rounded to 74%).

- What evidence do the authors have that the catenanes are the same structures in solution and the solid state? Are the relative 3D arrangements of the components the same (i.e. are components threaded through the same component 'entrances'? Given that the constituent cages have three 'entrances' to their cavities, there are several ways they could interlock to form a catenane (I think different threading would produce topological isomers).

Reply: The metallacages' framework features two potential threading entrances, as illustrated in Figure R1. Entrance 3, located between the naphthyridine groups, is too narrow for threading, with a width of 4.66 Å between the two nitrogen atoms of different naphthyridine groups (Figure R3b). Therefore, entrances 1 and 2 are suitable for the formation of catenanes. While the reviewer suggested that threading might occur at entrance 3, leading to significant differences in the 3D arrangement of these catenanes, the "window" between the two naphthyridine groups is too narrow for this to occur. Consequently, we propose that threading only occurs at entrances 1 and 2. However, these two entrances are indistinguishable in NMR spectra in solution. Furthermore, we have examined the structural isomers of both self-catenanes and cross-catenanes (Supplementary Figures 29 to 31). All possible relative isomers share a similar 3D arrangement, indicating that threading only occurs at two equal entrances. Therefore, we believe that the 3D arrangement of [2]catenanes in solution is similar to that observed in the solid state.

Figure R3. (a) entrances within the metallage **6**'s frameworks; (b) "window" size of entrance 3.

- The authors need to assign and explain changes in the $^1\text{H}/^{31}\text{P}$ NMR for the cross-catenanes vs the self catanenes – why do changes in chemical shifts occur?

Reply: Thank you for your valuable feedback. Chemical shifts and splitting are common in catenated systems, particularly due to the broken-symmetry of catenated SCCs as described in "*Nature* **367**, 720-723 (1994)" and shielding effects. The signals assignment in ^1H NMR spectra were done and the explanation ".....while the chemical shifts of **4+4** & **6+6** are occurred due to the temperature variation" & ".....and chemical shifts can be clearly observed due to the low-symmetric cross-catenated products" have been added to the manuscript. However, assigning the ^{31}P NMR signals to specific phosphines is challenging due to symmetry, and assigning the ^1H signals to specific protons in cross-catenane is nearly impossible due to the extremely low symmetry and interference from self-catenanes.

- Some figures are not well presented:

In figure 1, the crystal structures are hard for me to interpret (they are, in fact, better in the SI – figures S8 and S9 are really nice!).

In Figure 2. Why is the 9 from 8.9 a different font to the rest of the numbers

on the NMR scale?

Figure 4: Again, the crystal structures are near impossible to interpret. Also the use of cartoons that show non-interlocked rings is confusing and a little sloppy.

Reply: Sorry to confuse you, and all the figures have been modified as follow:

Figure 1 | Synthesis and single-crystal structures of metallacages and [2]catenanes.

(a) Synthesis of metallacage 4 and crystal structure of [2]catenane 4+4 (b) synthesis of metallacage 6 and crystal structures of [2]catenane 6+6 and cross-catenane 4+6. (Hydrogen atoms omitted for clarity)

Figure 2 | Varied temperature NMR experiments. (a) $^{31}\text{P}\{^1\text{H}\}$ NMR spectra (160 MHz) of 4 and 6 1:1 mixture in acetone- d_6 in different conditions (b) ^1H NMR spectra (400 MHz) of 4 and 6 mixtures in acetone- d_6 in different conditions.

Figure 4 | The global-minimum geometrical structures of [2]catenanes. (a) 4+4; (b) 4+6 and (c) 6+6 optimized at the B3LYP/def2-TZVP level of theory.

- The authors state: 'The cross-catenane 4+6 was thought to be the most thermodynamically favorable assembly among 4+4, 6+6, and 4+6, and the crystallization process was thermodynamically stable.' Clearly, this statement is not true for all conditions. Generally, the amount ('yield') of cross-catenane formed was small (and the reactions are under thermodynamic control). Please can the authors clarify what they mean. A discussion of entropy is also needed – presumably cross-catenation is entropically favoured (and hence why more cross-catenanes forms at higher temperatures).

Reply: We apologize for any confusion. The statement "The cross-catenane 4+6 was thought to be the most thermodynamically favorable assembly among 4+4, 6+6, and 4+6, and the crystallization process was thermodynamically stable" has been replaced with "To evaluate the stabilities and mechanisms of cross-catenation, ..." Additionally, a discussion on entropy

has been added to the manuscript, stating that “Meanwhile, the mathematical probability of the cross-catenation state (2/3) in a mixed system containing two metallacage **4** and two metallacage **6** is higher than that for the self-catenation state (1/3), indicating that the formation of cross-catenane **4+6** is entropically favorable in crystallization. This is because the minimal structural differences within the metallacages' frameworks strongly influence the hierarchical catenated assembly. Consequently, unique cross-catenated metallacages are preferentially formed in the crystal.”.

- The authors state ‘evidence for 8+8 [2]catenane was not observed’ but also present a well matched isotope pattern for 8+8 in figure 3c. This is confusing/contradictory – what are the authors trying to say?

Reply: The mass isotope pattern for **8+8** in Figure 3c is derived solely from the mass spectrum of **8**. We were unable to observe the isotope pattern for **8+8** in the mass spectrum of the **6 & 8** 1:1 mixture. We apologize for any confusion caused, and Figure 3c was intended to demonstrate the self-catenation of **8+8** in acetone solution with **8** alone. To eliminate any confusion, we have removed Figure 3c from the manuscript and added it to Supplementary Figure 6.

Reply to Reviewer #3

Comments: The paper is well written and the results are properly rationalised and justified. The work represents a nice example of synergic usage of experiments and modelling techniques in the formation of supramolecular metallocage systems. The work is of good significance in the the field and experimental and computational techniques under use are up to date. The work is original: it contains new metallocage systems. The work support the conclusions. Data data analysis, interpretation and conclusions are reasonable. The applied methodology is appropriate and respects the standards in the field. The details provided are enough to reproduce the results.

Reply: We greatly appreciate the reviewer’s very positive comments and useful advices. Herewith, we addressed the reviewer’s comments as follows:

I suggest to furnish also the geometries of the optimised assemblies obtained by DFT calculations.

Reply: Thank you for the comments. The geometrical coordinates of optimized assemblies obtained by DFT calculations have already been added in the supporting information (Supplementary Table 8~12) in the updated version.

I wish to suggest further references to the same subject:

J Martí-Rujas, S Ma, A Famulari - Inorganic Chemistry, 2022

J Martí-Rujas, S Elli, A Famulari - Scientific Reports, 2023

Reply: These two references have been cited as ref. 21 & ref. 22.

Reviewers' Comments:

Reviewer #1:

Remarks to the Author:

The authors have addressed all the issues raised by the reviewers. The current revised manuscript can be accepted for publication.

Reviewer #2:

Remarks to the Author:

The authors have largely addressed my concerns. I think the paper is substantially more accessible to the reader and would be well placed in Nature Communications.

The only outstanding query relates to the thermodynamic stability of the cross-catenane. I did not find the authors' reply satisfactory, because they do not consider why the stability of the cross-catenane is temperature dependent. It is notable that there is almost no cross-catenane present at low temperature but there is a lot at high temperature. Whilst rationalising this does not change the conclusions of the paper, I think it is an interesting observation that is worthy of more discussion. This would give insight into how to favour cross-catenation over self-catenation - which would be a nice extra facet to add to the work.

Reply to Reviewer #1

Comments: The authors have addressed all the issues raised by the reviewers. The current revised manuscript can be accepted for publication.

Reply: Thanks a lot for reviewer's recommendation.

Reply to Reviewer #2

Comments: The authors have largely addressed my concerns. I think the paper is substantially more accessible to the reader and would be well placed in Nature Communications.

Reply: We greatly appreciate the reviewer's useful advices and kind recommandation. Herewith, we addressed the reviewer's comments as follows:

The only outstanding query relates to the thermodynamic stability of the cross-catenane. I did not find the authors' reply satisfactory, because they do not consider why the stability of the cross-catenane is temperature dependent. It is notable that there is almost no cross-catenane present at low temperature but there is a lot at high temperature. Whilst rationalising this does not change the conclusions of the paper, I think it is an interesting observation that is worthy of more discussion. This would give insight into how to favour cross-catenation over self-catenation - which would be a nice extra facet to add to the work.

Reply: Thank you for your comments. Relational discussion "That is, heating disrupts the self-catenanes, resulting in the formation of more thermodynamically stable cross-catenanes." have been added to the manuscript.